# Investigations on Nonlinear Dynamic Modeling and Vibration Responses of T-Shaped Beam Structures

Shuai Chen [1], Dengqing Cao [1,*], Jin Wei [2], Guiqin He [1], Bo Fang [1] and Youxia Li [3]

1 School of Astronautics, Harbin Institute of Technology, Harbin 150001, China
2 School of Electromechanical and Automotive Engineering, Yantai University, Yantai 264005, China
3 China Academy of Space Technology, Beijing 110105, China
* Correspondence: dqcao@hit.edu.cn

**Abstract:** A novel nonlinear dynamic modeling approach is proposed for the T-shaped beam structures widely used in the field of aerospace. All of the geometrical nonlinearities including the terms in the deformation of the beams, the terms at the connections, and the free ends of beams are considered in the dynamic modeling process. The global mode method is employed to determine the natural frequencies and global mode shapes of the linearized system. The validity and accuracy of the derived model are verified by comparing the natural frequencies obtained with those calculated from FEM. Adopting the Galerkin truncation procedure, a set of reduced-order nonlinear ODEs is obtained for the structure. A study on the variation of dynamic responses taking the different numbers of global modes into account is performed to determine the number of modes taken in nonlinear vibration analysis. A comparison between the responses of the system with linear or nonlinear matching and boundary conditions is given to evaluate the importance of neglecting and reserving the nonlinear terms in matching and boundary conditions. It is shown that ignoring the nonlinear terms in both matching and boundary conditions may significantly alter the responses while developing the discretized governing ODEs of the structure.

**Keywords:** T-shaped beam structure; nonlinear dynamic modeling; global modal method; nonlinear dynamic behavior; nonlinear matching conditions

## 1. Introduction

A specific structure composed of multiple beams is usually used as a component in large-scale space structures, especially in the design of flexible spacecraft [1–4]. With the advancement of aerospace technology and the diversification of aerospace missions, the use of flexibility and large-scale structures has gradually become an important trend in the development of spacecraft, which leads to geometric nonlinearity as a factor that has to be considered. Hence, establishing accurate nonlinear dynamic models of such systems is an important basis for nonlinear dynamical analysis and active vibration control. This is of important practical significance for predicting and understanding the dynamic behavior of such flexible structures under the effect of applied loadings.

The nonlinear dynamic behavior of a single beam has been widely studied in the literature [5–11]. Compared with a single beam, multi-beam structures have more complex dynamic characteristics due to internal coupling and geometric and inertial nonlinearities. Haddow et al. [12] established the equation of motion for the L-shaped beam structure with only quadratic nonlinearities considering the influence of the axial movement of the beam due to bending and obtained the approximate solutions of the obtained equation by using the multiple scales method. Nayfeh et al. [13–17] conducted a comprehensive experimental and theoretical study on the nonlinear motions of the L-shaped beam structure, and the research showed that small excitation could produce chaotic motion under two-to-one internal resonance. Additionally, the theoretically deduced periodic, quasi-periodic and

chaotic responses are in good agreement with experimental results. Hamed Samandari et al. [18] obtained the nonlinear differential equations of motion for the L-shaped beam structure by using the Hamilton principle and Euler–Bernoulli beam theory, considering the large deformation of the structure. The differential quadrature method (DQM) is used to discretize the differential equations of motion in the space domain. The harmonic balance method is used to further transform the differential equations of motions into nonlinear algebraic equations for numerical solutions and numerical analysis of nonlinear responses. What is more, Kumar et al. [19] conducted an investigation of nonlinear phenomena of dynamic responses of N-link manipulators under 3:1 internal resonance considering geometric nonlinearities.

Dynamic models of the multi-beam structures are continuous with infinite degrees of freedom. To obtain a reduced-order dynamic model, the modal analysis method can be used to discretize the continuous systems. Therefore, the accuracy of the calculation results depends highly on the quality of the mode shapes used in the dynamic modeling. For a multi-beam structure, due to the interaction between the components, the mode function of each beam is different from that of a single beam. Consequently, the dynamical model based on the global mode has the advantages of lower dimensions and high precision, which provides convenience for dynamical analysis. The global mode method (GMM) was proposed to establish the dynamic model of the micro-electric static comb [20] and the composite flexible structure of a long-span cable-stayed bridge [21]. Then, the modal conversion and nonlinear dynamic response analysis were conducted. Wei et al. applied GMM to establish a spacecraft model with a deployable solar panel [22], a multi-beam structure with nonlinear hinges [23], and a nonlinear motion model of the L-shaped beam mass structure [24]. Based on these models, a series of inherent characteristic analysis and dynamic response studies were respectively performed. Considering the out-of-plane transverse deformation and torsional deformation of the beams, Cao et al. [25] established a linear dynamic model of the T-shaped beams and studied the vibration suppression based on piezoelectric plates.

For multi-beam structures connected by joints, there are many studies on the effect of joints on the dynamic behavior of the system. Vakil et al. [26] studied the effect on the inherent properties of the system due to changes in joint stiffness and determined the upper limit of joint stiffness, which can distinguish whether the joint of the manipulator is flexible or rigid. Meng et al. [27] investigated the effect of joint stiffness on the vibration response of the system. The research indicated that appropriate joint stiffness can reduce the overall vibration of the structure due to the coupling relationship between flexible rods and flexible joints. Recently, Wei et al. [23,28–30] derived a reduced-order analytical model for multi-beam structures connected with hinges associated with a nonlinear rotational spring. Based on this model, they investigated the effect of the nonlinear stiffness and damping of the joints on the attitude and position of a spacecraft during maneuvering. Studies had shown that the nonlinear stiffness of the joint has a great influence on the nonlinear response of the system.

Based on previous preliminary research [31], in this article, along with the idea of the GMM, a novel nonlinear dynamic modeling approach is proposed for the T-shaped beam. All of the geometrical nonlinearities including the terms in the deformation of the beams, the stress compatibility condition at the connections of beams, and the terms at the free ends are taken into account in the dynamic modeling process. The generalized Hamiltonian principle is employed to establish the nonlinear partial differential equations of motion for the T-shaped beam structure. Using the method proposed in [20–23], the natural frequency of the system and the corresponding global mode functions are worked out. Using the global mode functions and their orthogonality relations, an explicit set of reduced-order nonlinear ordinary differential equations of motion is obtained. Combined with specific examples, the precision and effectiveness of the model are verified by comparing the natural frequency and the global mode shape of the system. Through the dynamic equations given by the numerical examples, the dynamic responses of the system with

different numbers of modes are studied. The importance of reserving or neglecting the nonlinear terms in matching and boundary conditions when deriving the discrete control equations is evaluated.

## 2. Nonlinear Dynamic Model in the Continuous Form

Consider the in-plane motion of a T-shaped beam structure that is fixed on a horizontal moving base, as shown in Figure 1. The T-shaped beam is composed of three lightweight inextensible beams, namely a horizontal beam and two vertical beams. For the sake of description, the horizontal beam is called Beam-1, the lower vertical beam is Beam-2, and the upper vertical beam is Beam-3. Coordinate frame $o_1x_1y_1$ is a fixed inertial frame with the origin at the left end of the horizontal beam, and coordinate frames $o_2x_2y_2$ and $o_3x_3y_3$ are satellite inertial frames with the origins both at $(l_1, v_1(l_1, t))$. $l_i, \rho_i, E_i, I_i, A_i, u_i(x_i, t)$ and $v_i(x_i, t)$ denote the length, the density, Young's modulus, the inertia moment, the cross-sectional area, and the axial and lateral displacements of the $i$-th beam, respectively. Moreover, it is assumed that the shear deformation and warpage of all beams can be ignored and the beams are inextensible [32,33].

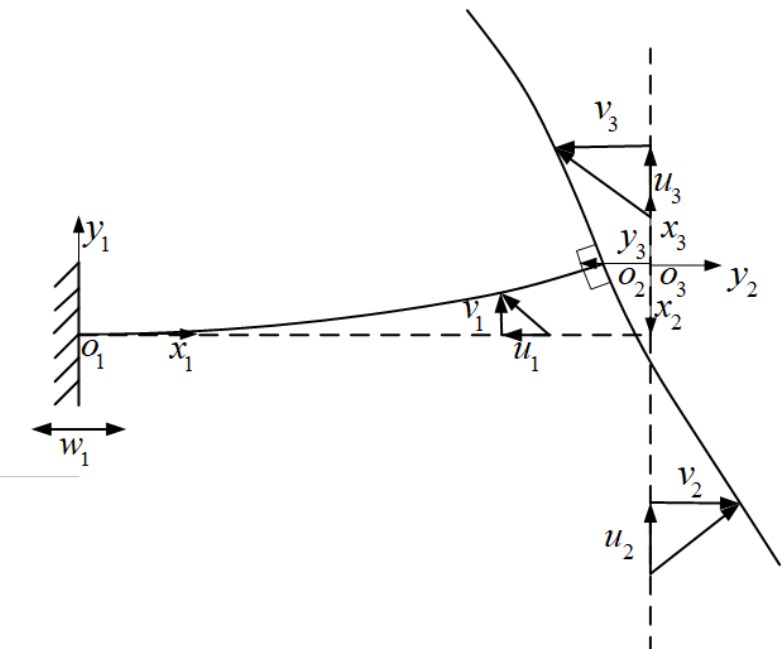

**Figure 1.** Schematic of T-shaped beam structure.

The generalized Hamilton's principle is expressed as

$$\int_{t_1}^{t_2} (\delta T - \delta V + \delta W) dt = 0, \tag{1}$$

where T, V and W represent the kinetic energy, potential energy and work done by the non-conservative force of the system, respectively.

Take a micro-element on the beam, as shown in Figure 2, where the dashed and solid lines indicate the positions of the micro-element before and after deformation, respectively. The symbol $\theta (x, t)$ in Figure 2 represents the angular displacement of the micro element. The in-extensibility condition can be depicted by the schematic $pp_1 = pp'_1$.

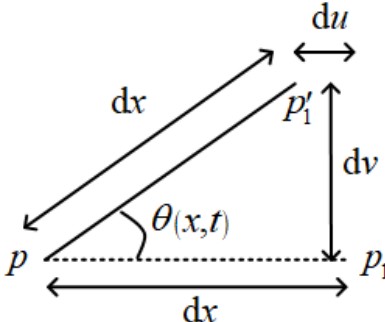

**Figure 2.** Schematic of the geometry and the in-extensibility condition.

The relations between these displacements are determined by the geometry. We write [34]

$$\theta_i(x_i, t) = arcsin\left[v_i'(x_i, t)\right], \tag{2}$$

$$cos[\theta_i(x_i, t)] = \frac{dx_i + du_i(x_i, t)}{dx_i} = 1 + u_i'(x_i, t), \tag{3}$$

where the superposed dot represents the partial derivative with respect to time, and (′) represents the partial derivative with respect to displacement, $i$ = 1, 2, 3. This notation is followed quite extensively from here on. Expanding Equations (2) and (3) yields

$$\theta_i(x_i, t) = v_i'(x_i, t) + \frac{1}{6}v_i'^3(x_i, t) + \cdots , \tag{4}$$

$$u_i'(x_i, t) = -\frac{1}{2}\theta_i^2(x_i, t) + \frac{1}{24}\theta_i^4(x_i, t) + \cdots . \tag{5}$$

Substituting Equation (4) into Equation (5) yields

$$u_i'(x_i, t) = -\frac{1}{2}v_i'^2(x_i, t) + O\left(v_i'^4(x_i, t)\right), \tag{6}$$

where $O\left(v_i'^4(x_i, t)\right)$ is neglected in the subsequent analysis. The axial displacement of any point on the beams can be obtained by integrating Equation (6) from the origin to this point on the beams,

$$u_i(x_i, t) = -\frac{1}{2}\int_0^{x_i} v_i'^2(y, t)dy. \tag{7}$$

In this way, $u_1(l_1, t)$ represents the axial displacement of the right end of the horizontal beam.

The curvature is expressed as

$$\kappa_i = \frac{\partial}{\partial x_i}\theta_i(x_i, t) = v_i''(x_i, t) + \frac{1}{2}v_i'^2(x_i, t)v_i''(x_i, t) + \cdots . \tag{8}$$

The potential energy of the T-shaped beam structure is given by

$$V = \frac{1}{2}\int_0^{l_1}\int_{A_1} E_1 z_1^2 \kappa_1^2 dA_1 dx_1 + \frac{1}{2}\int_0^{l_2}\int_{A_2} E_2 z_2^2 \kappa_2^2 dA_2 dx_2 + \frac{1}{2}\int_0^{l_3}\int_{A_3} E_3 z_3^2 \kappa_3^2 dA_3 dx_3. \tag{9}$$

Substitute Equation (8) into Equation (9) to obtain

$$\begin{aligned} V = &\frac{1}{2}\int_0^{l_1} E_1 I_1 v_1''^2(x_1, t)dx_1 + \frac{1}{2}\int_0^{l_1} E_1 I_1 \left[v_1'(x_1, t)v_1''(x_1, t)\right]^2 dx_1 \\ &+ \frac{1}{2}\int_0^{l_2} E_2 I_2 v_2''^2(x_2, t)dx_2 + \frac{1}{2}\int_0^{l_2} E_2 I_2 \left[v_2'(x_2, t)v_2''(x_2, t)\right]^2 dx_2 \\ &+ \frac{1}{2}\int_0^{l_3} E_3 I_3 v_3''^2(x_3, t)dx_3 + \frac{1}{2}\int_0^{l_3} E_3 I_3 \left[v_3'(x_3, t)v_3''(x_3, t)\right]^2 dx_3. \end{aligned} \tag{10}$$

The kinetic energy of the T-shaped beam structure is given by

$$
\begin{aligned}
T =& \tfrac{1}{2}\int_0^{l_1} \rho_1 \left\{ \left[ \dot{w}_s(t) + \dot{u}_1(x_1,t) \right]^2 + \dot{v}_1^{\,2}(x_1,t) \right\} dx_1 \\
&+ \tfrac{1}{2}\int_0^{l_2} \rho_2 \left\{ \left[ \dot{v}_1(l_1,t) - \dot{u}_2(x_2,t) \right]^2 + \left[ \dot{w}_s(t) + \dot{v}_2(x_2,t) \right]^2 \right\} dx_2 \\
&+ \tfrac{1}{2}\int_0^{l_3} \rho_3 \left\{ \left[ \dot{v}_1(l_1,t) + \dot{u}_3(x_3,t) \right]^2 + \left[ \dot{w}_s(t) - \dot{v}_3(x_3,t) \right]^2 \right\} dx_3.
\end{aligned}
\tag{11}
$$

Substituting Equation (7) into Equation (11) yields

$$
\begin{aligned}
T =& \tfrac{1}{2}\int_0^{l_1} \rho_1 \left\{ \left[ \dot{w}_s(t) - \int_0^{x_1} \tfrac{1}{2}\tfrac{\partial}{\partial t}\left( \tfrac{\partial v_1(y,t)}{\partial y} \right)^2 dy \right]^2 + \left( \tfrac{\partial v_1(x_1,t)}{\partial t} \right)^2 \right\} dx_1 \\
&+ \tfrac{1}{2}\int_0^{l_2} \rho_2 \left\{ \left[ \tfrac{\partial v_1(l_1,t)}{\partial t} + \int_0^{x_2} \tfrac{1}{2}\tfrac{\partial}{\partial t}\left( \tfrac{\partial v_2(y,t)}{\partial y} \right)^2 dy \right]^2 + \left[ \dot{w}_s(t) + \tfrac{\partial v_2(x_2,t)}{\partial t} \right]^2 \right\} dx_2 \\
&+ \tfrac{1}{2}\int_0^{l_3} \rho_3 \left\{ \left[ \tfrac{\partial v_1(l_1,t)}{\partial t} - \int_0^{x_3} \tfrac{1}{2}\tfrac{\partial}{\partial t}\left( \tfrac{\partial v_3(y,t)}{\partial y} \right)^2 dy \right]^2 + \left[ \dot{w}_s(t) - \tfrac{\partial v_3(x_3,t)}{\partial t} \right]^2 \right\} dx_3.
\end{aligned}
\tag{12}
$$

Considering the viscous damping in the structure, the virtual work done by the non-conservative forces is given by

$$
\begin{aligned}
\delta W =& -\int_0^{l_1} c\dot{v}_1(x_1,t)\delta v_1(x_1,t)dx_1 - \int_0^{l_2} c\dot{v}_2(x_2,t)\delta v_2(x_2,t)dx_2 \\
& -\int_0^{l_3} c\dot{v}_3(x_3,t)\delta v_3(x_3,t)dx_3.
\end{aligned}
\tag{13}
$$

Substituting Equations (10), (12), and (13) into Equation (1) and simplifying, we obtain the nonlinear vibration equations of the structure as

$$
\begin{aligned}
&\rho_1\ddot{v}_1(x_1,t) + c\dot{v}_1(x_1,t) + E_1 I_1 \left\{ v_1''''(x_1,t) + \left[ v_1'(x_1,t)\left[ v_1'(x_1,t)v_1''(x_1,t) \right]' \right]' \right\} \\
&+ \tfrac{1}{2}\rho_1 \left[ v_1'\int_{l_1}^{x_1}\int_0^\theta \tfrac{\partial^2}{\partial t^2}\left[ v_1'^{\,2}(y,t) \right]dyd\theta \right]' - \rho_1 v_1'(x_1,t)\ddot{w}_s(t) + \left[ \rho_1\ddot{w}_s(t)(l_1 - x_1) \right. \\
&\left. + E_2 I_2 v_2'''(0,t) - E_3 I_3 v_3'''(0,t) - (\rho_2 l_2 + \rho_3 l_3)v_2'(0,t)\ddot{v}_1(l_1,t) \right]v_1''(x_1,t) = 0,
\end{aligned}
\tag{14}
$$

$$
\begin{aligned}
&\rho_2\ddot{v}_2(x_2,t) + c\dot{v}_2(x_2,t) + E_2 I_2 \left\{ v_2''''(x_2,t) + \left[ v_2'(x_2,t)\left[ v_2'(x_2,t)v_2''(x_2,t) \right]' \right]' \right\} \\
&+ \tfrac{1}{2}\rho_2 \left[ v_2'(x_2,t)\int_{l_2}^{x_2}\int_0^\theta \tfrac{\partial^2}{\partial t^2}\left[ v_2'^{\,2}(x_2,t) \right]dyd\theta \right]' + \rho_2 v_2'(x_2,t)\ddot{v}_1(l_1,t) \\
&- \rho_2(l_2 - x_2)v_2''(x_2,t)\ddot{v}_1(l_1,t) + \rho_2\ddot{w}_s(t) = 0,
\end{aligned}
\tag{15}
$$

$$
\begin{aligned}
&\rho_3\ddot{v}_3(x_3,t) + c\dot{v}_3(x_3,t) + E_3 I_3 \left\{ v_3''''(x_3,t) + \left[ v_3'(x_3,t)\left[ v_3'(x_3,t)v_3''(x_3,t) \right]' \right]' \right\} \\
&+ \tfrac{1}{2}\rho_3 \left[ v_3'(x_3,t)\int_{l_3}^{x_3}\int_0^\theta \tfrac{\partial^2}{\partial t^2}\left[ v_3'^{\,2}(x_3,t) \right]dyd\theta \right]' - \rho_3 v_3'(x_3,t)\ddot{v}_1(l_1,t) \\
&+ \rho_3(l_3 - x_3)v_3''(x_3,t)\ddot{v}_1(l_1,t) - \rho_3\ddot{w}_s(t) = 0.
\end{aligned}
\tag{16}
$$

The boundary conditions of the T-shaped beam structure are expressed as
at $x_1 = 0$

$$
v_1(0,t) = 0,\ v_1'(0,t) = 0,
\tag{17}
$$

at $x_2 = l_2$

$$
E_2 I_2 v_2''(l_2,t) + E_2 I_2 v_2'^{\,2}(l_2,t)v_2''(l_2,t) = 0,
\tag{18}
$$

$$
E_2 I_2 \left[ v_2'''(l_2,t) + v_2'(l_2,t)v_2''^{\,2}(l_2,t) + v_2'^{\,2}(l_2,t)v_2'''(l_2,t) \right] = 0,
\tag{19}
$$

at $x_3 = l_3$

$$
E_3 I_3 v_3''(l_3,t) + E_3 I_3 v_3'^{\,2}(l_3,t)v_3''(l_3,t) = 0,
\tag{20}
$$

$$
E_3 I_3 \left[ v_3'''(l_3,t) + v_3'(l_3,t)v_3''^{\,2}(l_3,t) + v_3'^{\,2}(l_3,t)v_3'''(l_3,t) \right] = 0,
\tag{21}
$$

where Equations (18)–(21) have not been simplified in order to preserve the completeness of the nonlinearity of the boundary conditions.

At the connection interfaces of the beams, the displacement and rotation angle of the horizontal beam and the vertical beams must match, and the bending moments and forces must be balanced. The matching conditions at the junction are expressed as

$$v_2(0,t) = -v_3(0,t) = u_1(l_1,t),$$ (22)

$$v_1'(l_1,t) + \frac{1}{6}v_1'^3(l_1,t) = v_2'(0,t) + \frac{1}{6}v_2'^3(0,t) = v_3'(0,t) + \frac{1}{6}v_3'^3(0,t),$$ (23)

$$E_1 I_1 \left[v_1''(l_1,t) + v_1'^2(l_1,t)v_1''(l_1,t)\right] - \frac{2+v_1'^2(l_1,t)}{2+v_2'^2(0,t)} \cdot E_2 I_2 \left[v_2''(0,t) + v_2'^2(0,t)v_2''(0,t)\right]$$
$$-\frac{2+v_1'^2(l_1,t)}{2+v_3'^2(0,t)} \cdot E_3 I_3 \left[v_3''(0,t) + v_3'^2(0,t)v_3''(0,t)\right] = 0,$$ (24)

$$E_1 I_1 \left[v_1'''(l_1,t) + v_1'(l_1,t)v_1''^2(l_1,t) + v_1'^2(l_1,t)v_1'''(l_1,t)\right]$$
$$-\rho_2 l_2 \ddot{v}_1(l_1,t) - \rho_2 \int_0^{l_2} \int_0^{x_2} \frac{1}{2}\frac{\partial^2}{\partial t^2}(v_2'^2(y,t))dydx_2$$
$$-\rho_3 l_3 \ddot{v}_1(l_1,t) + \rho_3 \int_0^{l_3} \int_0^{x_3} \frac{1}{2}\frac{\partial^2}{\partial t^2}(v_3'^2(y,t))dydx_3$$
$$+E_2 I_2 v_1'(l_1,t)v_2'''(0,t) - \rho_2 l_2 v_1'(l_1,t)v_2'(0,t)\ddot{v}_1(l_1,t)$$
$$-E_3 I_3 v_1'(l_1,t)v_3'''(0,t) - \rho_3 l_3 v_1'(l_1,t)v_3'(0,t)\ddot{v}_1(l_1,t) = 0,$$ (25)

$$-E_2 I_2 \left[v_2'''(0,t) + v_2'(0,t)v_2''^2(0,t) + v_2'^2(0,t)v_2'''(0,t)\right]$$
$$+\rho_2 v_2'(0,t) \cdot \int_0^{l_2} \int_0^{x_2} \frac{1}{2}\frac{\partial^2}{\partial t^2}\left(\frac{\partial v_2(y,t)}{\partial y}\right)^2 dydx_2 + \rho_2 l_2 v_2'(0,t)\frac{\partial^2 v_1(l_1,t)}{\partial t^2}$$
$$+E_3 I_3 \left[v_3'''(0,t) + v_3'(0,t)v_3''^2(0,t) + v_3'^2(0,t)v_3'''(0,t)\right]$$
$$+\rho_3 l_3 v_3'(0,t) \cdot \frac{\partial^2 v_1(l_1,t)}{\partial t^2} - \rho_3 v_3'(0,t) \cdot \int_0^{l_3} \int_0^{x_3} \frac{1}{2}\frac{\partial^2}{\partial t^2}\left(\frac{\partial v_3(y,t)}{\partial y}\right)^2 dydx_3$$
$$+E_1 I_1 v_1'(l_1,t)v_1'''(l_1,t) = 0.$$ (26)

where the Equations (24)–(26) are the matching equations of the bending moment, the vertical force and the horizontal force at the junction, respectively.

## 3. Natural Characteristics of the System

The natural frequencies and global mode functions of the system can be obtained by linearizing the nonlinear partial differential equations of motion for the structure. Ignoring the nonlinear terms and the damping terms in Equations (14)–(16) yields the linear partial differential equations of the transverse motion of the *i*-th beam

$$E_i I_i v_i''''(x_i,t) + \rho_i \ddot{v}_i(x_i,t) = 0,$$ (27)

Similarly, ignoring the nonlinear terms in the boundary and matching conditions (17)–(26), we have

$$\begin{aligned}
&v_1(0,t) = 0, \, v_1'(0,t) = 0, \\
&E_2 I_2 v_2''(l_2,t) = 0, \, E_2 I_2 v_2'''(l_2,t) = 0, \\
&E_3 I_3 v_3''(l_3,t) = 0, \, E_3 I_3 v_3'''(l_3,t) = 0, \\
&v_2(0,t) = 0, \, v_3(0,t) = 0, \\
&v_1'(l_1,t) = v_2'(0,t) = v_3'(0,t), \\
&E_1 I_1 v_1'''(l_1,t) = (\rho_2 l_2 + \rho_3 l_3)\ddot{v}_1(l_1,t), \\
&E_1 I_1 v_1''(l_1,t) - E_3 I_3 v_3''(0,t) = E_2 I_2 v_2''(0,t).
\end{aligned}$$ (28)

Assume that the displacements $v_i(x_i,t)$ are separable in space and time,

$$v_i(x_i,t) = \varphi_i(x_i)e^{j\omega t},$$ (29)

where $\omega$ is the natural frequency of the system.

Substituting Equation (29) into the linear vibration Equation (27) of the beam yields

$$E_i I_i \varphi_i''''(x_i) - \omega^2 \rho_i \varphi_i(x_i) = 0.$$ (30)

The general solution of Equation (30) can be written as

$$\varphi_i(x_i) = B_i cos(\beta_i x_i) + C_i sin(\beta_i x_i) + D_i cosh(\beta_i x_i) + G_i sinh(\beta_i x_i), \ x_i \in [0, l_i], \tag{31}$$

where $\beta_i = \left(\rho_i \omega^2 / E_i I_i\right)^{1/4}$.

The constants $B_i, C_i, D_i, G_i, (i = 1 \sim 4)$ should be determined using the boundary and matching conditions given in Equation (28). Substituting expression (31) into Equation (28), the eigenvalue equation for the linearized system of the T-shaped beam is obtained, and its matrix form is expressed as follows

$$H(\omega)\Psi = 0, \tag{32}$$

where $\Psi = \begin{bmatrix} B_1 & C_1 & D_1 & G_1 & B_2 & C_2 & D_2 & G_2 & B_3 & C_3 & D_3 & G_3 \end{bmatrix}^T$, and entries of matrix $H(\omega) \in R^{12 \times 12}$ are given in Appendix A.

The positive roots of the frequency equation $|H(\omega)| = 0$, denoted in ascending order by $\omega_1, \omega_2, \omega_3, \cdots$, are the natural frequencies of the T-shaped beam structure. The eigenvector $\Psi^{(r)}$, where $r = 1, 2, 3, \cdots$, corresponding to the natural frequency $\omega_r$, can be obtained from Equation (32). Further, the $r$-th mode shapes for the T-shaped structure can be determined from Equation (31).

Referring to the derivation process in Reference [21], the orthogonality conditions of the global mode $\Phi(x) = \begin{bmatrix} \varphi_1(x) & \varphi_2(x) & \varphi_3(x) \end{bmatrix}^T$ with respect to mass and stiffness can be obtained, which are respectively expressed as

$$\int_0^{l_1} \rho_1 \varphi_{1r}(x)\varphi_{1s}(x)dx + \int_0^{l_2} \rho_2 \varphi_{2r}(x)\varphi_{2s}(x)dx + \int_0^{l_3} \rho_3 \varphi_{3r}(x)\varphi_{3s}(x)dx \\ + (\rho_2 l_2 + \rho_3 l_3)\varphi_{1r}(l_1)\varphi_{1s}(l_1) = M_s \delta_{rs}, \tag{33}$$

And

$$\int_0^{l_1} E_1 I_1 \varphi''_{1r}(x)\varphi''_{1s}(x)dx + \int_0^{l_2} E_2 I_2 \varphi''_{2r}(x)\varphi''_{2s}(x)dx \\ + \int_0^{l_3} E_3 I_3 \varphi''_{3r}(x)\varphi''_{3s}(x)dx = K_s \delta_{rs}, \tag{34}$$

where $M_s$ and $K_s$ are positive constants and $\delta_{rs}$ is the Kronecker delta.

## 4. Nonlinear Dynamic Model in the Discrete Form

The nonlinear dynamic model in the continuous form described by Equations (14)–(16) combined with the nonlinear boundary conditions (17)–(21) and the nonlinear matching conditions (22)–(26) can be simplified to a nonlinear dynamic model in the discrete form by using the global mode functions and their orthogonality relations presented in Section 3. The transverse displacements $v_i$ can be expressed in the following form

$$v_1 = \sum_{j=1}^{n} \varphi_{1j}(x_1)q_j(t), \ v_2 = \sum_{j=1}^{n} \varphi_{2j}(x_2)q_j(t), \ v_3 = \sum_{j=1}^{n} \varphi_{3j}(x_3)q_j(t), \tag{35}$$

where $\varphi_{1j}(x_1)$, $\varphi_{2j}(x_2)$ and $\varphi_{3j}(x_3)$ are the global mode functions for the T-shaped beam structure obtained from Equation (31), respectively, and $q_j(t)$ is the generalized coordinate for the whole system.

Substituting Equation (35) into Equations (14)–(16) yields

$$
\begin{aligned}
&\sum_{j=1}^{n} \rho_1 \varphi_{1j}(x_1)\ddot{q}_j + \sum_{j=1}^{n} c\varphi_{1j}(x_1)\dot{q}_j + \sum_{j=1}^{n} E_1 I_1 \varphi_{1j}''''(x_1)q_j \\
&+ \sum_{i,j,k=1}^{n} \tfrac{1}{2}\rho_1 \left( \varphi_{1i}'(x_1)q_i \int_{l_1}^{x_1} \int_0^\theta \varphi_{1j}'(y)\varphi_{1k}'(y)\tfrac{d^2}{dt^2}(q_j q_k)\,dy\,d\theta \right)' \\
&+ \sum_{i,j,k=1}^{n} E_1 I_1 \left( \varphi_{1i}'(x_1)\left[\varphi_{1j}'(x_1)\varphi_{1k}''(x_1)\right]' \right)' q_i q_j q_k \\
&- \sum_{i=1}^{n} \rho_1 \ddot{w}_s(t) q_i \left[ \varphi_{1i}'(x_1) + (x_1 - l_1)\varphi_{1i}''(x_1) \right] \\
&+ \sum_{i,j=1}^{n} \varphi_{1i}''(x_1)q_i \left[ +E_2 I_2 \varphi_{2j}'''(0)q_j - E_3 I_3 \varphi_{3j}'''(0)q_j \right] \\
&+ \sum_{i,j,k=1}^{n} \left[ \left( -\rho_3 l_3 \varphi_{3j}'(0) - \rho_2 l_2 \varphi_{2j}'(0) \right)\varphi_{1k}(l_1)q_j \ddot{q}_k \right] q_i \varphi_{1i}''(x_1) = 0,
\end{aligned}
\tag{36}
$$

$$
\begin{aligned}
&\sum_{j=1}^{n} \rho_2 \varphi_{2j}(x_2)\ddot{q}_j + \sum_{j=1}^{n} c\varphi_{2j}(x_2)\dot{q}_j + \sum_{j=1}^{n} E_2 I_2 \varphi_{2j}''''(x_2)q_j \\
&+ \sum_{i,j,k=1}^{n} E_2 I_2 \left( \varphi_{2i}'(x_2)\left[\varphi_{2j}'(x_2)\varphi_{2k}''(x_2)\right]' \right)' q_i q_j q_k + \rho_2 \ddot{w}_s(t) \\
&+ \sum_{i,j,k=1}^{n} \tfrac{1}{2}\rho_2 \left( \varphi_{2i}'(x_2)q_i \int_{l_2}^{x_2} \int_0^\theta \varphi_{2j}'(y)\varphi_{2k}'(y)\tfrac{d^2}{dt^2}(q_j q_k)\,dy\,d\theta \right)' \\
&+ \sum_{i,j=1}^{n} \rho_2 \varphi_{1i}(l_1)\ddot{q}_i q_j \left[ \varphi_{2j}'(x_2) + (x_2 - l_2)\varphi_{2j}''(x_2) \right] = 0,
\end{aligned}
\tag{37}
$$

$$
\begin{aligned}
&\sum_{j=1}^{n} \rho_3 \varphi_{3j}(x_3)\ddot{q}_j + \sum_{j=1}^{n} c\varphi_{3j}(x_3)\dot{q}_j + \sum_{j=1}^{n} E_3 I_3 \varphi_{3j}''''(x_3)q_j(t) \\
&+ \sum_{i,j,k=1}^{n} E_3 I_3 \left( \varphi_{3i}'(x_3)\left[\varphi_{3j}'(x_3)\varphi_{3k}''(x_3)\right]' \right)' q_i q_j q_k - \rho_3 \ddot{w}_s(t) \\
&+ \sum_{i,j,k=1}^{n} \tfrac{1}{2}\rho_3 \left( \varphi_{3i}'(x_3)q_i \int_{l_3}^{x_3} \int_0^\theta \varphi_{3j}'(y)\varphi_{3k}'(y)\tfrac{d^2}{dt^2}(q_j q_k)\,dy\,d\theta \right)' \\
&- \sum_{i,j=1}^{n} \rho_3 \varphi_{1i}(l_1)\ddot{q}_i q_j \left[ \varphi_{3j}'(x_3) + (x_3 - l_3)\varphi_{3j}''(x_3) \right] = 0.
\end{aligned}
\tag{38}
$$

Multiply both sides of Equations (36)–(38) by $\varphi_{1s}(x_1)$, $\varphi_{2s}(x_2)$ and $\varphi_{3s}(x_3)$, respectively, integrating them along the respective beam lengths. Then, the ordinary differential equations of motion for the T-shaped beam structure can be obtained by adding all resulting equations and simplifying by using matching and boundary conditions (17)–(26) and orthogonality relations (33), (34), namely,

$$
\begin{aligned}
&M_s \ddot{q}_s + K_s q_s + \sum_{j=1}^{n} \mu_s^j \dot{q}_j + \sum_{j=1}^{n} a_s^j \ddot{w}_s(t) q_j + \sum_{j,k=1}^{n} b_s^{jk} q_j \ddot{q}_k + \sum_{j,k=1}^{n} c_s^{jk} q_j q_k \\
&+ \sum_{j,k=1}^{n} d_s^{jk} \dot{q}_j \dot{q}_k + \sum_{j,k,r=1}^{n} e_s^{jkr} q_j q_k q_r + \sum_{j,k,r=1}^{n} h_s^{jkr} \dot{q}_j \dot{q}_k q_r + \sum_{j,k,r=1}^{n} p_s^{jkr} q_j q_k \ddot{q}_r \\
&= -\int_0^{l_2} \rho_2 \ddot{w}_s(t)\varphi_{2s}(x_2)\,dx_2 + \int_0^{l_3} \rho_3 \ddot{w}_s(t)\varphi_{3s}(x_3)\,dx_3, \quad s = 1, 2, \cdots, n.
\end{aligned}
\tag{39}
$$

where $\mu_s^j$ are constants that can be determined by the global mode functions and damping coefficient, and $a_s^j, b_s^{jk}, c_s^{jk}, d_s^{jk}, e_s^{jkr}, h_s^{jkr}, p_s^{jkr}$ given in Appendix B are constants that can be determined by the global mode functions. It is worth noting that the underlined terms in the expressions of those constants are from the non-linear terms in matching and boundary conditions. Therefore, the ordinary differential equations of motion under linear matching and boundary conditions, which are called the "incomplete nonlinear dynamic model (INDM)" hereafter, can be obtained by ignoring the underlined terms. Consequently, the ordinary differential Equation (39) with all of the geometrical and inertial nonlinear terms is called the "complete nonlinear dynamic model (CNDM)".

### 5. Model Validation for the Linearized System

In this section, a comparison of the natural frequencies and the global mode functions obtained theoretically with the results from commercial software ANSYS is performed by using a typical example to validate the approach proposed in this paper.

Now, a simple example of the T-shaped beam structure is provided. Assume that the material for all beams is steel with density $\rho$ = 7850 kg/m$^3$, Young's modulus $E$ = 200 $GPa$, damping ratio $c$ = 0.02, and Poisson's ratio $v$ = 0.31. The cross sections of all beams are $b$ = 0.012 m, $h$ = 0.002 m. The lengths of the beams are $l_1$ = 0.3 m, $l_2$ = 0.3 m, $l_3$ = 0.2 m, respectively.

Use the approach proposed to obtain the natural characteristics of the T-shaped beam structure, including natural frequencies and global mode shapes. Table 1 shows the natural frequencies of the T-shaped beam structure, taking the finite element results from ANSYS as a reference. The maximum relative error between the natural frequencies obtained by the current method and those from FEM is 0.1192%. The results show that the approach proposed in this paper is effective and the frequency obtained has higher accuracy because there is no approximation and neglection in the mathematical derivation of this method. Define the relative error as

$$Re^{(i)} = \left| \frac{\omega_{GMM,i} - \omega_{FEM,i}}{\omega_{FEM,i}} \right| \times 100\%. \ (i = 1, 2, \cdots, 8), \tag{40}$$

where $\omega_{GMM,i}$ and $\omega_{FEM,i}$ are the natural frequencies calculated by the global modal method and FEM, respectively.

**Table 1.** First 8 order frequency of T-shaped beam structure (rad/s).

| Frequency Order | Natural Frequency (GMM) | Natural Frequency (FEM) | Re (%) |
|---|---|---|---|
| 1 | 32.03 | 32.03 | 0.0000 |
| 2 | 94.20 | 94.20 | 0.0000 |
| 3 | 199.45 | 199.45 | 0.0000 |
| 4 | 579.64 | 579.46 | 0.0325 |
| 5 | 737.52 | 737.27 | 0.0341 |
| 6 | 1363.76 | 1362.89 | 0.0645 |
| 7 | 1846.45 | 1844.25 | 0.1192 |
| 8 | 2015.75 | 2014.12 | 0.0811 |

Figure 3 shows the first 8 modes obtained here and those from FEM, where the mode pictures obtained by FEM are drawn in the ORIGIN software using the data of ANSYS software. It can be seen that the calculation results of the two methods are in good agreement. The reduced-order nonlinear differential equations of motion of the T-shaped beam structure obtained by using the global modes are accurate.

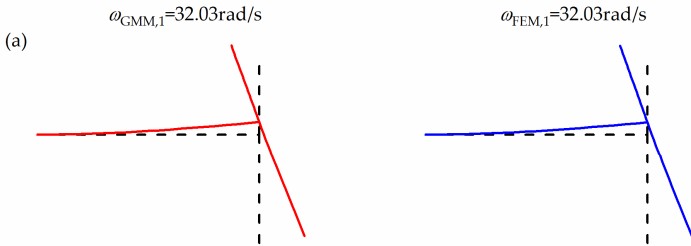

**Figure 3.** *Cont.*

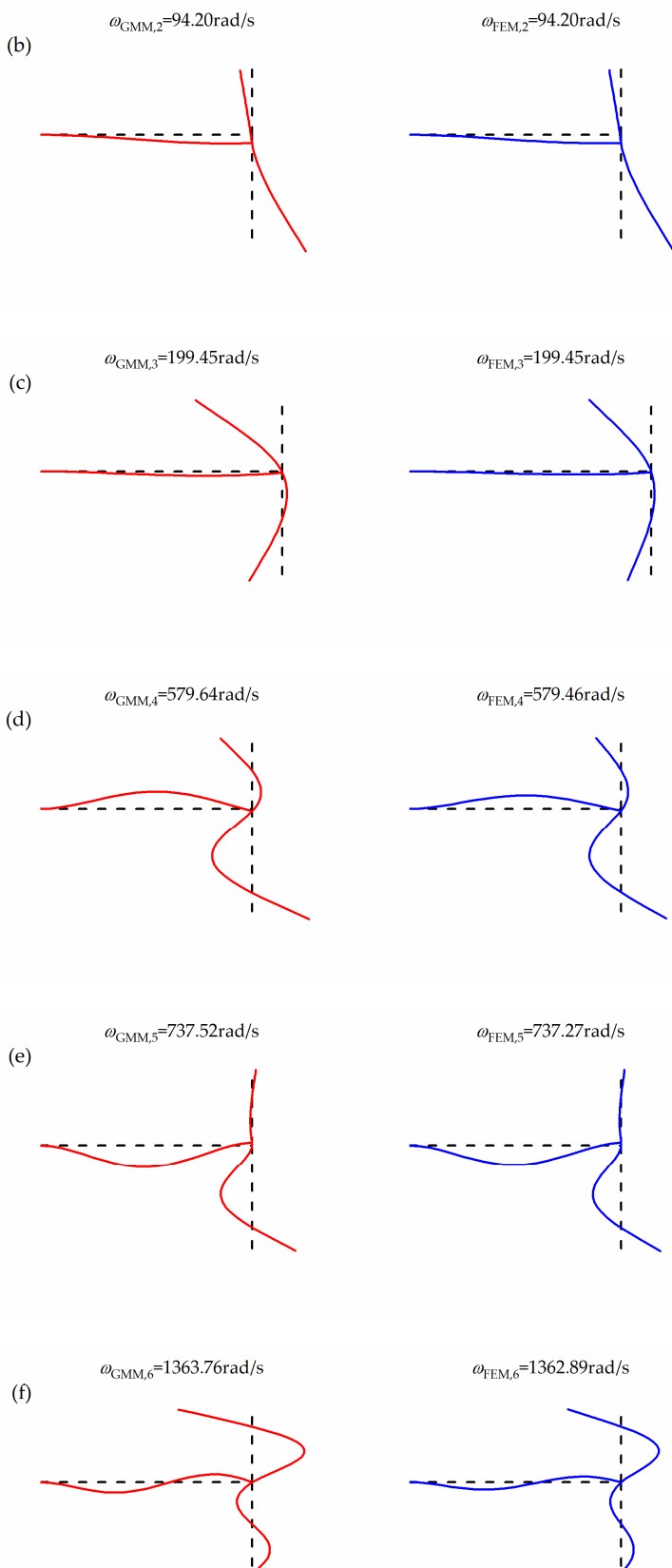

**Figure 3.** *Cont.*

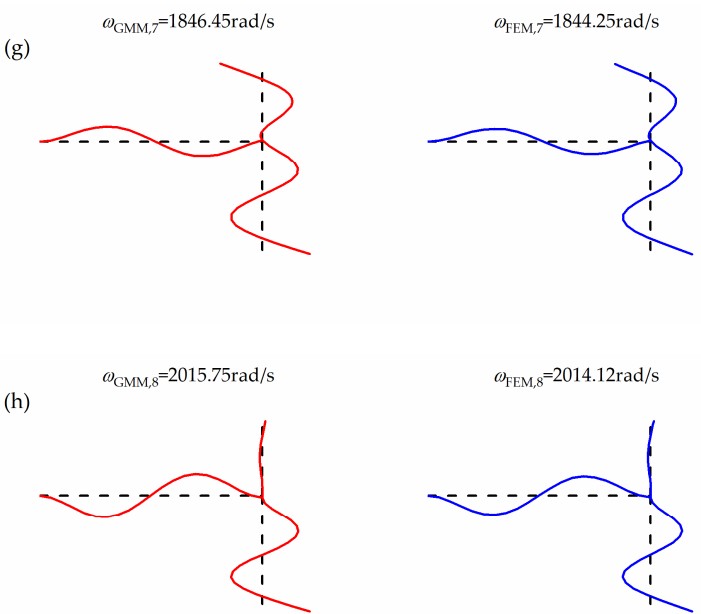

**Figure 3.** First 8 order modal shapes: GMM (Left column) and FEM (Right column).

## 6. Nonlinear Dynamical Responses and Discussions

The T-shaped beam structure is assumed to be fixed on a horizontal moving base, as shown in Figure 1. The displacement of the moving support is assumed as

$$w_s(t) = w_0 cos(\Omega t), \tag{41}$$

where $w_0$ is a constant and denotes the amplitude of the displacement, and $\Omega$ is the frequency of the moving base in rad/s.

In order to determine the number of modes that should be taken in vibration analysis, the dynamical responses of the linear dynamic model (without all of the nonlinear terms), the INDM, and the CNDM with $n$ modes under the sweeping frequency are given in Figures 4 and 5a,b, respectively, where the excitation amplitude $w_0$ = 0.0008 m, and the amplitudes of the responses are taken from the transverse displacements of the free end of Beam 2. It can be seen from Figure 4 that in the linear dynamic model, the resonance peaks of the system with the first $n$ modes ($n = 1, 2, \cdots, 6$) all appear at 32.03 rad/s; that is approximately equal to the first natural frequency, which illustrates that low-order frequencies play a leading role in vibration. There is a sudden increase in response amplitude of the T-shaped beam when the number of modes increases from 3 to 4; this may be because the coupling effect between the 4th mode and the first 3 modes is strong. Moreover, whether the system is a linear dynamic model, INDM or CNDM, the variation trend of the response value of the system with the first $n$ modes is consistent.

It can be observed from Figures 4 and 5 that 4 or more than 4 modes should be truncated to guarantee the solution accuracy of the system in the primary resonance region. This implies that the first 4 modes should be taken for simulation in the calculations of the nonlinear vibration responses.

The steady-state response time histories of Beam-2 end point under the excitation amplitude $w_0$ = 0.01 m and frequency $\Omega$ = 32.34 rad/s are calculated using the proposed method and the FEM, respectively, as shown in Figure 6. The maximum relative error between the displacement amplitudes from the proposed and finite element methods, defined by $\left( u - u^{FEM} \right) / u^{FEM}$, is 9.01%. Similarly, the maximum relative error between the velocity amplitudes from the proposed and finite element methods is 9.43%. It shows that the numerical results obtained by our method are matched very well with those from ANSYS software, which further confirms the effectiveness of the proposed method.

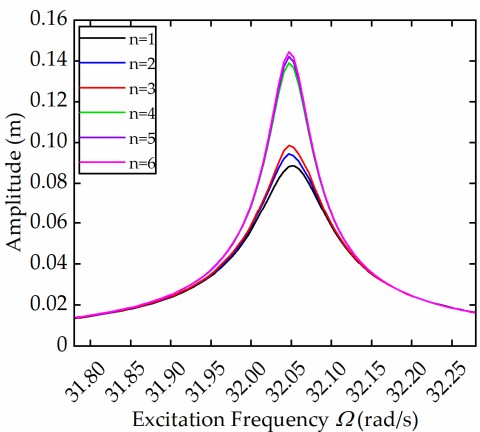

**Figure 4.** Frequency responses of the linear dynamic model with the first n modes.

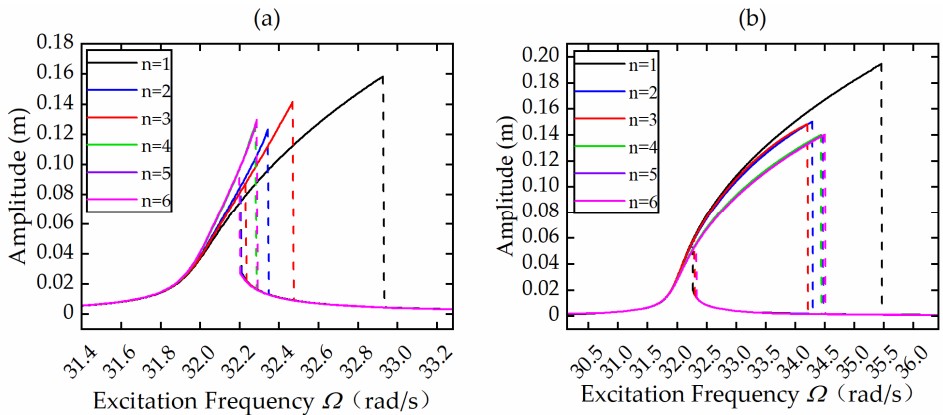

**Figure 5.** Frequency response curves of the system with the first n modes under different conditions: (**a**) INDM; (**b**) CNDM.

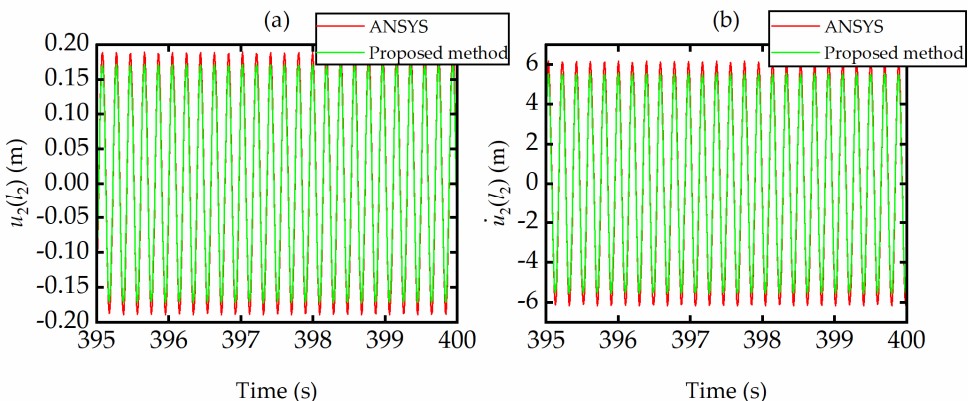

**Figure 6.** The steady−state response time histories of Beam−2 end point: (**a**) displacement; (**b**) velocity.

The force response curves of the INDM with the first *n* modes for the excitation frequency $\Omega = 32.15$ rad/s are shown in Figure 7a, and those of the CNDM with the first *n* modes for the excitation frequency $\Omega = 32.66$ rad/s are shown in Figure 7b. The same jumping phenomenon can be seen in the figure, and as the excitation amplitude increases, the difference in the dynamic response of both systems with the first *n* modes becomes larger and larger. When the vibration amplitude of the system is relatively large, more modes need to be adopted to meet the accuracy requirements. All of these clearly show

that the influence of higher-order modes on the system dynamic response is closely related to the excitation amplitude of the system.

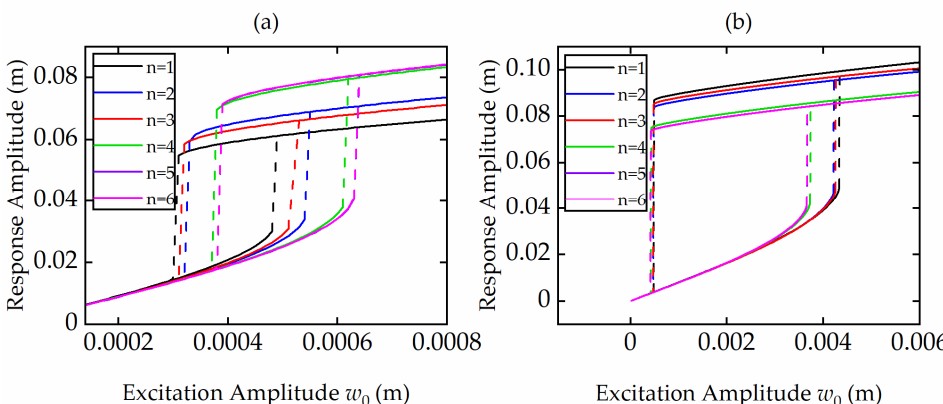

**Figure 7.** Amplitude response curves of the INDM (**a**) and the CNDM (**b**) with the first n modes.

The influences of excitation amplitudes on the responses of the nonlinear system for the INDM and the CNDM are shown in Figure 8a,b respectively. The curves show that the responses are positively correlated with the excitation amplitude in both systems. It can also be seen that there is a jump phenomenon in the nonlinear response. The greater the excitation, the more obvious the jump, and the higher the corresponding jump frequency.

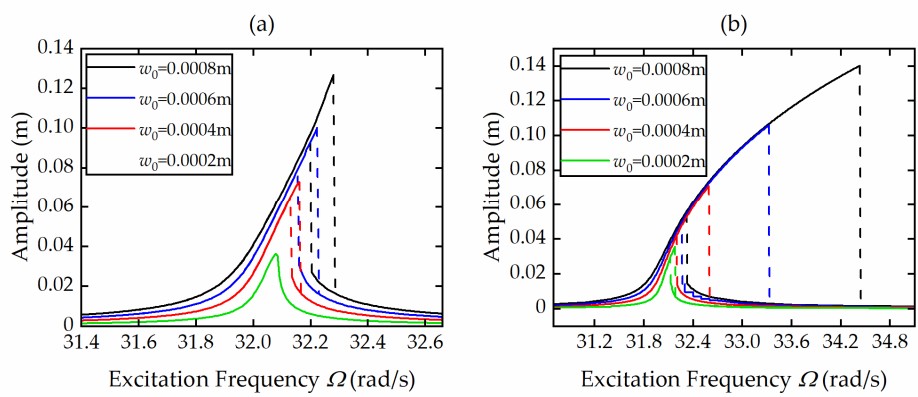

**Figure 8.** Frequency responses of the INDM (**a**) and the CNDM (**b**) with different excitation amplitudes.

To evaluate the importance of reserving and ignoring nonlinear matching and boundary conditions when deriving discrete nonlinear governing equations, the dynamic responses of the nonlinear systems INDM and CNDM at the excitation amplitude $w_0 = 0.0008$ m, $w_0 = 0.0006$ m, $w_0 = 0.0004$ m and $w_0 = 0.0002$ m are shown in Figure 9a–d, respectively. The curves show that there is a jumping phenomenon in both systems. As the excitation amplitude increases, the response gap between the two systems gradually increases, the jumping frequency of the system CNDM is much higher, and the hysteresis zone is wider in comparison with that of the system INDM. Therefore, when developing the discretized governing equations of temporal modes under a certain degree of excitation amplitude, we should not only consider the nonlinear terms in the vibration differential equations of beams but also the nonlinear terms in the matching and boundary conditions. Ignoring the nonlinear terms in the matching and boundary conditions may lead to relatively large errors in the dynamic responses of the system.

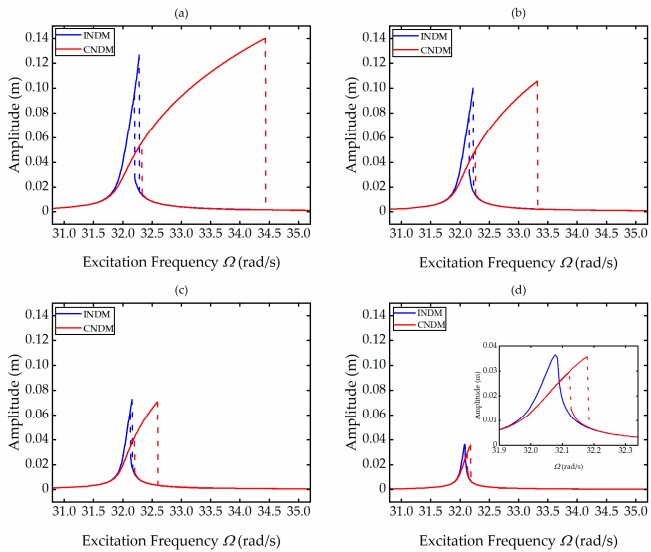

**Figure 9.** Frequency responses of the INDM and the CNDM at different excitation amplitudes: (**a**) $w_0 = 0.0008$ m; (**b**) $w_0 = 0.0006$ m; (**c**) $w_0 = 0.0004$ m; (**d**) $w_0 = 0.0002$ m.

Figures 10–12 respectively show the steady-state responses of a system containing second-order and third-order nonlinear terms, second-order nonlinear terms only, and third-order nonlinear terms only, when the excitation frequency is near the first-order natural frequency, where the response is the transverse vibration of the free end of Beam 2, including time history of steady state, phase portrait, Poincare map, and spectrum response. The Poincare map is a point composed of integer multiples of the excitation period in all three pictures, which means that the vibration of the T-shaped beam structure reaches a periodic stable state. In the spectrogram (d) of both Figures 10 and 11, peaks appear at zero, one, two, and three times the fundamental frequency. In the spectrogram (d) of Figure 12, peaks only appear at one and three times the fundamental frequency. It can be inferred that the peaks at three times the fundamental frequency are caused by the second and third nonlinear terms, while the peaks at zero and two times the fundamental frequency are mainly caused by the second-order nonlinear terms in the system.

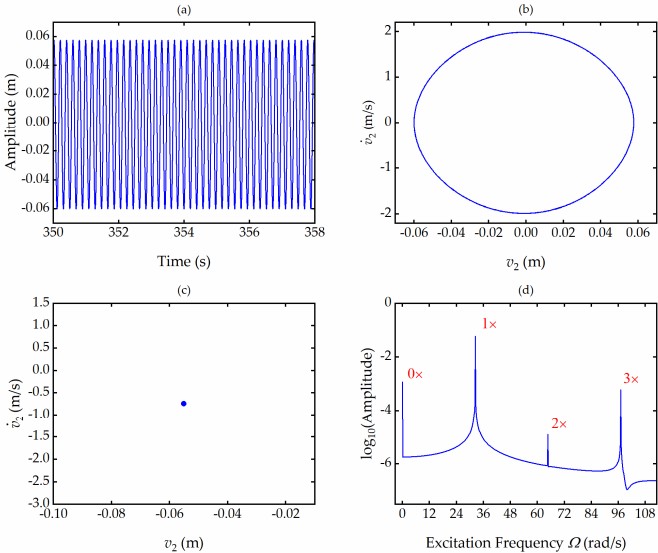

**Figure 10.** Response of a system with second− and third−order nonlinear terms for the case of $w_0 = 0.0008$ m, $\Omega = 32.34$ rad/s: (**a**) Time history of steady state. (**b**) Phase portrait. (**c**) Poincare map. (**d**) Spectrum response.

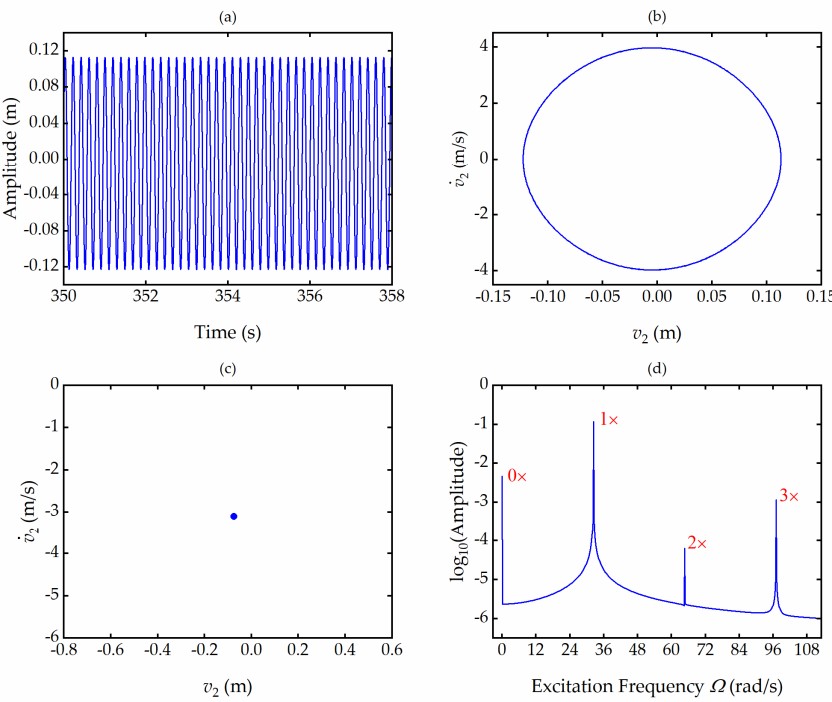

**Figure 11.** Response of a system with only second−order nonlinear terms for the case of $w_0 = 0.0008$ m, $\Omega = 32.34$ rad/s: (**a**) Time history of steady state. (**b**) Phase portrait. (**c**) Poincare map. (**d**) Spectrum response.

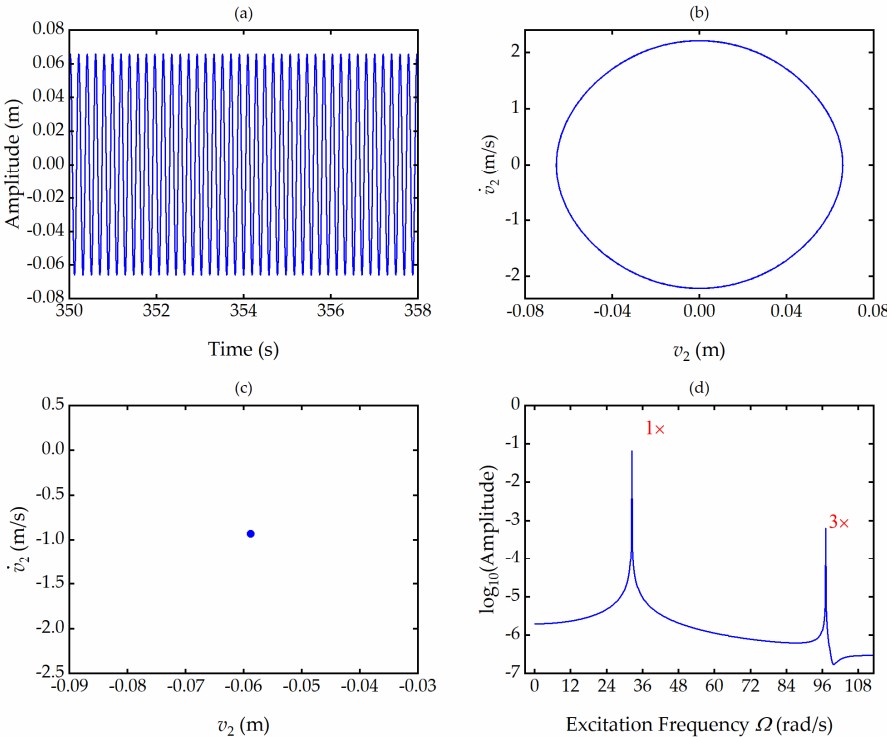

**Figure 12.** Response of a system with only third−order nonlinear terms for the case of $w_0 = 0.0008$ m, $\Omega = 32.34$ rad/s: (**a**) Time history of steady state. (**b**) Phase portrait. (**c**) Poincare map. (**d**) Spectrum response.

## 7. Conclusions

A novel nonlinear dynamic modeling approach has been presented for the T-shaped beam, in which all of the geometrical nonlinearities including the terms in the deformation of the beams, in the stress compatibility condition at the connections of beams, and the terms at the free ends are taken into account in the dynamic modeling process. The nonlinear ordinary differential equations of motion describing the dynamic behaviors of the T-shaped beam structure have been established, and a series of dynamic response analyses have been performed. Considering the geometric nonlinearity of the beam, and assuming that the beam is not extensible, the partial differential equations of motion of the T-shaped beam structure are obtained by using the generalized Hamiltonian principle, along with their nonlinear matching and boundary conditions. The global mode method was employed to obtain the natural frequency of the system and the corresponding global mode functions. The validity of the dynamic model obtained was verified by comparing the natural frequencies obtained by the proposed approach with the calculation results of FEM. The Galerkin truncation procedure was employed to obtain the nonlinear ordinary differential equations of motion with a lower degree of freedom. To study the effect of the nonlinear terms in the boundary and matching conditions on the dynamical responses, the numerical solutions for the CNDM and the INDM were obtained, respectively, and a comparison of the results was given. Conclusions drawn from the dynamic modeling procedure and discussions on the nonlinear vibration responses are as follows:

1.  The dynamical responses are dominated by the low-order modes of the system. The numerical example shows that the first 4 modes should be taken for simulation in the calculations of the nonlinear vibration responses. In order to improve the calculation efficiency, fewer modes should be selected for calculation. Moreover, the responses of the nonlinear dynamic systems have a strong dependence on the excitation amplitude.
2.  Ignoring the nonlinear terms in the matching and boundary conditions may reduce the accuracy of the system. In CNDM, nonlinearity is continuous along the structure; owing to the nonlinear terms of the vibration equations, boundary and matching conditions are all reserved. Contrarily, there are breakpoints of nonlinearity at the junction and boundaries in INDM, where the nonlinear terms of the boundary and matching conditions are neglected, which may lead to unacceptable errors of the dynamic responses. Therefore, when studying the nonlinear dynamic response of a multi-beam structure, the nonlinear terms in the boundary and matching conditions should be reserved and are indeed worthy of our attention.

**Author Contributions:** Conceptualization, S.C. and Y.L.; methodology, S.C. and D.C.; validation, G.H. and B.F.; data curation, G.H.; writing—original draft preparation, S.C.; writing—review and editing, J.W.; funding acquisition, D.C. and Y.L. All authors have read and agreed to the published version of the manuscript.

**Funding:** This research was funded by the National Key Technology R&D Program of China, grant number 2020YFB1506702-03, the Civil Space Technology Advance Research Project of the Administration of Science, Technology and Industry, grant number D020201, and the National Natural Science Foundation of China, grant number 11732005.

**Data Availability Statement:** Not applicable.

**Acknowledgments:** The authors gratefully acknowledge the support of DQ research group, Institute of Aerospace Vehicle Dynamics and Control, Harbin Institute of Technology.

**Conflicts of Interest:** The authors declare no conflict of interest.

## Appendix A

Entries of the matrix $H(\omega)$ in Equation (32) are

$$
\begin{aligned}
& H_{11} = H_{13} = H_{22} = H_{24} = 1, \\
& H_{12} = H_{14} = H_{15} = H_{16} = H_{17} = H_{18} = H_{19} = H_{110} = H_{111} = H_{112} = 0, \\
& H_{21} = H_{23} = H_{25} = H_{26} = H_{27} = H_{28} = H_{29} = H_{210} = H_{211} = H_{212} = 0, \\
& H_{31} = H_{32} = H_{33} = H_{34} = H_{39} = H_{310} = H_{311} = H_{312} = 0, \\
& H_{35} = -cos(\beta_2 l_2), H_{36} = -sin(\beta_2 l_2), H_{37} = cosh(\beta_2 l_2), H_{38} = sinh(\beta_2 l_2), \\
& H_{41} = H_{42} = H_{43} = H_{44} = H_{49} = H_{410} = H_{411} = H_{412} = 0, \\
& H_{45} = sin(\beta_2 l_2), H_{46} = -cos(\beta_2 l_2), H_{47} = sinh(\beta_2 l_2), H_{48} = cosh(\beta_2 l_2), \\
& H_{51} = H_{52} = H_{53} = H_{54} = H_{55} = H_{56} = H_{57} = H_{58} = 0, \\
& H_{59} = -cos(\beta_3 l_3), H_{510} = -sin(\beta_3 l_3), H_{511} = cosh(\beta_3 l_3), H_{512} = sinh(\beta_3 l_3), \\
& H_{61} = H_{62} = H_{63} = H_{64} = H_{65} = H_{66} = H_{67} = H_{68} = 0, \\
& H_{69} = sin(\beta_3 l_3), H_{610} = -cos(\beta_3 l_3), H_{611} = sinh(\beta_3 l_3), H_{612} = cosh(\beta_3 l_3), \\
& H_{71} = H_{72} = H_{73} = H_{74} = H_{76} = H_{78} = H_{79} = H_{710} = H_{711} = H_{712} = 0, \\
& H_{75} = H_{77} = H_{89} = H_{811} = H_{96} = H_{98} = 1, \\
& H_{81} = H_{82} = H_{83} = H_{84} = H_{85} = H_{86} = H_{87} = H_{88} = H_{810} = H_{812} = 0, \\
& H_{91} = H_{92} = H_{93} = H_{94} = H_{95} = H_{97} = H_{99} = H_{911} = 0, \\
& H_{910} = H_{912} = H_{106} = H_{108} = -1, \\
& H_{101} = -sin(\beta_1 l_1), H_{102} = cos(\beta_1 l_1), H_{103} = sinh(\beta_1 l_1), H_{104} = cosh(\beta_1 l_1), \\
& H_{105} = H_{107} = H_{109} = H_{1010} = H_{1011} = H_{1012} = 0, \\
& H_{1101} = sin(\beta_1 l_1) + \beta_1(l_2 + l_3)cos(\beta_1 l_1), H_{1102} = -cos(\beta_1 l_1) + \beta_1(l_2 + l_3)sin(\beta_1 l_1), \\
& H_{1103} = sinh(\beta_1 l_1) + \beta_1(l_2 + l_3)cosh(\beta_1 l_1), H_{1104} = cosh(\beta_1 l_1) + \beta_1(l_2 + l_3)sinh(\beta_1 l_1), \\
& H_{1105} = H_{1106} = H_{1107} = H_{1108} = H_{1109} = H_{1110} = H_{1111} = H_{1112} = 0, \\
& H_{1201} = -cos(\beta_1 l_1), H_{1202} = -sin(\beta_1 l_1), H_{1203} = cosh(\beta_1 l_1), H_{1204} = sinh(\beta_1 l_1), \\
& H_{1205} = H_{1209} = 1, H_{1206} = H_{1208} = H_{1210} = H_{1212} = 0, H_{1207} = H_{1211} = -1.
\end{aligned} \tag{A1}
$$

## Appendix B

The constants in Equation (39) are

$$
\begin{aligned}
& \mu_s^j = c \sum_{i=1}^{3} \int_0^{l_i} \varphi_{ij}(x_i) \varphi_{is}(x_i) dx_i; \\
& a_s^j = -\rho_1 \int_0^{l_1} \varphi_{1s}(x_1) \left[ \varphi_{1j}'(x_1) + (x_1 - l_1)\varphi_{1j}''(x_1) \right] dx_1; \\
& b_s^{jk} = \rho_2 \varphi_{1k}(l_1) \int_0^{l_2} \varphi_{2s}(x_2) \left[ \varphi_{2j}'(x_2) + (x_2 - l_2)\varphi_{2j}''(x_2) \right] dx_2 + \rho_2 \varphi_{1s}(l_1) \int_0^{l_2} \int_0^{x_2} \varphi_{2k}'(y) \varphi_{2j}'(y) dy dx_2 \\
& \quad -\rho_3 \varphi_{1k}(l_1) \int_0^{l_3} \varphi_{3s}(x_3) \left[ \varphi_{3j}'(x_3) + (x_3 - l_3)\varphi_{3j}''(x_3) \right] dx_3 - \rho_3 \varphi_{1s}(l_1) \int_0^{l_3} \int_0^{x_3} \varphi_{3k}'(y) \varphi_{3j}'(y) dy dx_3 \\
& \quad +\tfrac{1}{2} \int_0^{l_1} {\varphi_{1s}'}^2(x_1) dx_1 \times \varphi_{1k}(l_1) \left[ \rho_2 l_2 \varphi_{2j}'(0) + \rho_3 l_3 \varphi_{3j}'(0) \right]; \\
& c_s^{jk} = \left[ E_2 I_2 \varphi_{2j}'''(0) + E_3 I_3 \varphi_{3j}'''(0) \right] \int_0^{l_1} \varphi_{1s}(x_1) \varphi_{1k}''(x_1) dx_1 - \varphi_{1s}(l_1) \varphi_{1k}'(l_1) \left[ E_2 I_2 \varphi_{2j}'''(0) - E_3 I_3 \varphi_{3j}'''(0) \right] \\
& \quad -\tfrac{1}{2} \int_0^{l_1} {\varphi_{1s}'}^2(x_1) dx_1 \times E_1 I_1 \varphi_{1j}'(l_1) \varphi_{1k}'''(l_1); \\
& d_s^{jk} = \rho_2 \varphi_{1s}(l_1) \int_0^{l_2} \int_0^{x_2} \varphi_{2k}'(y) \varphi_{2j}'(y) dy dx_2 - \rho_3 \varphi_{1s}(l_1) \int_0^{l_3} \int_0^{x_3} \varphi_{3k}'(y) \varphi_{3j}'(y) dy dx_3;
\end{aligned} \tag{A2}
$$

$$
\begin{aligned}
e_s^{jkr} &= \sum_{i=1}^{3} E_i I_i \int_0^{l_i} \varphi_{is}(x_i) \left( \varphi'_{ir}(x_i) \left[ \varphi'_{ij}(x_i) \varphi''_{ik}(x_i) \right]' \right)' dx_i \\
&+ \left[ E_1 I_1 \varphi'_{1r}(l_1) \varphi'_{1j}(l_1) \varphi''_{1k}(l_1) - E_2 I_2 \varphi'_{2r}(0) \varphi'_{2j}(0) \varphi''_{2k}(0) - E_3 I_3 \varphi'_{3r}(0) \varphi'_{3j}(0) \varphi''_{3k}(0) \right] \varphi'_{1s}(l_1) \\
&- E_1 I_1 \varphi_{1s}(l_1) \left[ \varphi'_{1r}(l_1) \varphi''_{1j}(l_1) \varphi''_{1k}(l_1) + \varphi'_{1r}(l_1) \varphi'_{1j}(l_1) \varphi'''_{1k}(l_1) \right] \\
&+ E_2 I_2 \varphi'_{2s}(l_2) \varphi'_{2j}(l_2) \varphi'_{2k}(l_2) \varphi''_{2r}(l_2) + E_3 I_3 \varphi'_{3s}(l_3) \varphi'_{3j}(l_3) \varphi'_{3k}(l_3) \varphi''_{3r}(l_3) \\
&- E_2 I_2 \varphi_{2s}(l_2) \left[ \varphi'_{2r}(l_2) \varphi''_{2j}(l_2) \varphi''_{2k}(l_2) + \varphi'_{2r}(l_2) \varphi'_{2j}(l_2) \varphi'''_{2k}(l_2) \right] \\
&- E_3 I_3 \varphi_{3s}(l_3) \left[ \varphi'_{3r}(l_3) \varphi''_{3j}(l_3) \varphi''_{3k}(l_3) + \varphi'_{3r}(l_3) \varphi'_{3j}(l_3) \varphi'''_{3k}(l_3) \right] \\
&- \frac{1}{2} \int_0^{l_1} \varphi'^2_{1s}(x_1) dx_1 \times E_2 I_2 \left[ \varphi'_{2j}(0) \varphi''_{2k}(0) \varphi''_{2r}(0) + \varphi'_{2j}(0) \varphi'_{2k}(0) \varphi'''_{2r}(0) \right] \\
&+ \frac{1}{2} \int_0^{l_1} \varphi'^2_{1s}(x_1) dx_1 \times E_3 I_3 \left[ \varphi'_{3j}(0) \varphi''_{3k}(0) \varphi''_{3r}(0) + \varphi'_{3j}(0) \varphi'_{3k}(0) \varphi'''_{3r}(0) \right] ; \\
h_s^{jkr} &= \sum_{i=1}^{3} \rho_i \int_0^{l_i} \varphi_{is}(x_i) \left( \varphi'_{ir}(x_i) \int_{l_i}^{x_i} \int_0^{\theta} \varphi'_{ij}(y) \varphi'_{ik}(y) dy d\theta \right)' dx_i \\
&- \frac{1}{2} \int_0^{l_1} \varphi'^2_{1s}(x_1) dx_1 \times \left[ -\rho_2 \varphi'_{2r}(0) \int_0^{l_2} \int_0^{x_2} \varphi'_{2j}(x_2) \varphi'_{2k}(x_2) dy dx_2 + \rho_3 \varphi'_{3r}(0) \int_0^{l_3} \int_0^{x_3} \varphi'_{3j}(x_3) \varphi'_{3k}(x_3) dy dx_3 \right] ; \\
p_s^{jkr} &= \left( -\rho_3 l_3 \varphi'_{3j}(0) - \rho_2 l_2 \varphi'_{2j}(0) \right) \varphi_{1r}(l_1) \int_0^{l_1} \varphi_{1s}(x_1) \varphi''_{1k}(x_1) dx_1 + \sum_{i=1}^{3} \rho_i \int_0^{l_i} \varphi_{is}(x_i) \left( \varphi'_{ik}(x_i) \int_{l_i}^{x_i} \int_0^{\theta} \varphi'_{ir}(y) \varphi'_{ij}(y) dy d\theta \right)' dx_i \\
&+ \varphi_{1s}(l_1) \varphi_{1r}(l_1) \varphi'_{1j}(l_1) \left[ \rho_2 l_2 \varphi'_{2k}(0) + \rho_3 l_3 \varphi'_{3k}(0) \right] + \frac{1}{2} \int_0^{l_1} \varphi'^2_{1s}(x_1) dx_1 \times \rho_2 \varphi'_{2j}(0) \int_0^{l_2} \int_0^{x_2} \varphi'_{2k}(x_2) \varphi'_{2r}(x_2) dy dx_2 \\
&- \frac{1}{2} \int_0^{l_1} \varphi'^2_{1s}(x_1) dx_1 \times \rho_3 \varphi'_{3j}(0) \int_0^{l_3} \int_0^{x_3} \varphi'_{3k}(x_3) \varphi'_{3r}(x_3) dy dx_3 .
\end{aligned}
\tag{A3}
$$

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
