# Peer review of "Investigations on Nonlinear Dynamic Modeling and Vibration Responses of T-Shaped Beam Structures"

_actuators, doi:10.3390/act11100293_

Round 1

Reviewer 1 Report

A study on the nonlinear dynamics of a T-shaped beam structure under base excitation is presented.

The structure is modeled considering inextensible beams and neglecting the shear deformation.

Natural frequencies and mode shapes are shown, and a comparison with a FEM analysis is given and discussed.

The nonlinear equations of motion are discretized using the Galerkin approach, and results are given in terms of frequency-response curves, time histories, phase-planes, Poincaré maps, and spectra.

The authors put emphasis on neglecting or retaining nonlinear terms within the boundary conditions.

The following questions arise while reading the manuscript:

- missing reference sources in the Introduction,

- section 2 is complitely free of any citations,

- formulae (4, 5, 8); the meaning of the is missing,

- formulae (4, 5, 8); please, check if the formulae hold for I=1;

- formula (8), please, check the correctness of the constant term L. There should be no constant terms after the partial derivative.

In conclusion the analysis is not clear enough, and since some formulae need to be checked carefully, the manuscript deserves major revisions.

Best regards.

Reviewer 2 Report

This paper focuses in T-shaped beam structures. A nonlinear dynamic modeling is proposed. A comparison betwem linear and nonlinear model is made.

This paper is interesting even if the authors published an similar article for L-shape beam structures. The weakness of the pape ris the lack of experimental investigation. It must be added. This investigation is essential to judge the efficiency of the model.

Conclusions : What are the perspectives ?

Reviewer 3 Report

In the work entitled “Investigations on Nonlinear Dynamic Modeling and Vibration 2 Responses of T-shaped Beam Structures” are studied the planar vibrations of T-shaped beam structures including geometrical nonlinearities. The Faedo-Galerkin method is adopted to discretize the system and frequency response curves (FRC) are determined for periodic excitations due to a harmonic motion of the beam support. Studies on the effect of the maximum base displacement amplitude are performed.

General comments:

The novelty of the paper is unclear. The mechanical model adopted is known and adopted in the literature since forever. Effects of the truncation of the Faedo-Galerkin discretization are studied in a multitude of beam-like structures since decades. The analyses reported in this work (i.e., the dynamic response to periodic base excitation) and the phenomena investigated (i.e., hardening response of geometrically nonlinear beams and the effect of the motion amplitude) are both very well known in the literature.

The authors are invited to check the huge literature available on this topic and to discuss it properly in order to, eventually, highlight the novelty of the work. Moreover, self-citations cover almost one third of the references and must be reduced. Among the others, the Reviewer suggests the work https://doi.org/10.2514/2.2054. Even the practical applications should be conveniently considered, such as nonlinearities due to structural damages, see for instance https://www.scopus.com/inward/record.uri?eid=2-s2.0-84994438007&partnerID=40&md5=167ec83efc63df8e15f4e4be489a9c7a.

The English used in this work is inappropriate and must be largely revised not only in terms of grammar and syntax but also in terms of the scientific formalism adopted.

This reviewer knows very well the mechanic formulation of geometrically nonlinear beams, the authors are kindly invited to check the equations reported in the manuscript since some of those need to be reformulated. For instance:

Eq. (2) is wrong, the flexural rotational angle theta is NOT the arcsine of the first derivative of the flexural displacement v. The correct relationship is \theta=arctan[v’/(1+u’)], being u the axial displacement of the beam. If the authors would have calculated the shear strain and considering unshearability, they would have calculated the correct expression of \theta.

Eq. (6) may be formally correct, but it is NOT obtained in the way the authors showed in the manuscript! It can be obtained by calculating the axial nonlinear stretch and then, considering the unstretchability.

Eq. (23) does not represent compatibility of flexural rotations since the authors are, in this work, the authors are considering NONLINEAR rotations as reported in Eq. (4), there is no sense in mixing up linear and nonlinear formulations without a proper motivation.

Please, be aware that \Omega is always used as the symbol to represent a circular frequency (rad/s), not a frequency (Hz=1/s). Therefore Eq. (51) bust be corrected.

The details of the calculation of the mode shapes Eq. (33) are, nowadays, trivial and can be avoided. The same considerations, in general, hold for the large part of Section 3 which can be shorten up.

This reviewer suggests to represent the y-axis of the FFT in Figs. 9-10-11 (d) in log scale.

Results shown in Figs. 9-10-11 (a,b,c) are trivial

Authors are invited to submit a widely revised version of the manuscript to be reviewed again; the Reviewer’s decision is “Major revision”.

Please, be aware that, if in the revised version the authors will not have improved the work (replying to all points highlighted by this reviewer), will not have clarified the novelties of the work and will not have improved the English of the text, the work will not be accepted.

Reviewer 4 Report

A novel nonlinear dynamic modeling approach has been presented for the T-shaped beam, in which all the geometrical nonlinearities including the terms in the deformation of the beams, in the boundary conditions are taken into account in the dynamic modeling process. The nonlinear ordinary differential equations of motion describing the dynamic behaviors of the T-shaped beam structure have been established and a series of dynamic responses analysis have been performed.  However, the paper should be carefully revised to make a concise demostration.

1. The manuscript should be carefully checked since symbol "?" appeared in many places of the manuscript. All the references and some symols in the equations are not identified.

2. Some symbols in the equation should be explained for easy reading.

3. Legend should be marked for the coordinate axis of figures. 

4. The author is suggested to give an explaination on the sudden increase of response amplitudes of the T-shaped beam when the number of the modes increases from 3 to 4.

5. In the introduction, the references mentioned by the authors are all about L-shaped beam. What about the ones about T-shaped beam?

6. The response analyzed with the proposed method should be compared to the ones with other methods.  The comparisons will make the proposed method more convinced.

Round 2

Reviewer 1 Report

The manuscript has been revised accordingly to the suggestions.

Thus, it can be accepted for publication.

Best regards.

Reviewer 2 Report

Due to the deadline for manuscript revision, the authors were unable to complete the corresponding experiment.  However, they add the finite element simulation results to verify the proposed method. 

The point-to-point responses to the comments are satisfactory. 

Reviewer 3 Report

The authors fulfilled the requests of this reviewer. The work can be accepted